# Mendelian randomization analysis using mixture models for robust and efficient estimation of causal effects

Guanghao Qi[1] & Nilanjan Chatterjee [1,2]

Mendelian randomization (MR) has emerged as a major tool for the investigation of causal relationship among traits, utilizing results from large-scale genome-wide association studies. Bias due to horizontal pleiotropy, however, remains a major concern. We propose a novel approach for robust and efficient MR analysis using large number of genetic instruments, based on a novel spike-detection algorithm under a normal-mixture model for underlying effect-size distributions. Simulations show that the new method, MRMix, provides nearly unbiased or/and less biased estimates of causal effects compared to alternative methods and can achieve higher efficiency than comparably robust estimators. Application of MRMix to publicly available datasets leads to notable observations, including identification of causal effects of BMI and age-at-menarche on the risk of breast cancer; no causal effect of HDL and triglycerides on the risk of coronary artery disease; a strong detrimental effect of BMI on the risk of major depressive disorder.

---

[1] Department of Biostatistics, Bloomberg School of Public Health, Johns Hopkins University, Baltimore, MD 21205, USA. [2] Department of Oncology, School of Medicine, Johns Hopkins University, Baltimore, MD 21205, USA. Correspondence and requests for materials should be addressed to N.C. (email: nilanjan@jhu.edu)

Discoveries of genetic susceptibility variants underlying complex traits continue to increase rapidly with ever-growing size of genome-wide association studies[1–5]. Mendelian randomization (MR)—a form of instrumental variable analysis for the assessment of the causal effect of one trait on another—provides major opportunity for translation of the increasing knowledge of genetics to improve human health[6,7]. MR analysis has already been widely used for obtaining evidence for drug targets, causal basis for epidemiologic associations, and cascading effects among complex molecular traits[8,9]. While MR was originally designed to be used with individual genetic instruments with known biologic functions, the recent trend has been to exploit the multitude of genetic variants emerging from large GWAS. The use of multiple genetic variants can allow to increase the power of MR analysis, correct for weak instrument bias and can provide evidence of causality across a broader set of underlying mechanisms for intervening on the traits[7,8].

When all of the selected variants satisfy the key assumption of MR analysis, i.e., they only have direct effects on one trait, then the causal effect of that trait on the other can be efficiently estimated by meta-analysis of the well-known ratio estimates across the different variants[10]. It is, however, recognized that while the availability of many variants provides an opportunity to strengthen MR analysis, there is potential for bias as the key assumption, i.e., the variants have no direct effect on the second trait, can be violated due to pleiotropy[7,8]. Indeed, recent empirical studies[3,11–16] have unequivocally shown that common variants have wide spread pleiotropic effects, and consequently, polygenic MR analysis can be susceptible to bias. Originally, some methods were developed that allow genetic variants to have pleiotropic effects, but they require the strong InSIDE assumption, i.e. the direct and indirect effects are uncorrelated[16–19]. Under this assumption, any genetic correlation between the two traits, as measured by the selected SNPs, can arise solely due to the underlying causal relationship between the traits.

The effects of genetic variants on multiple traits can be correlated when they are mediated through common causal factors. Thus, most recently there has been effort to develop methods that can allow for the presence of invalid instruments which may have complex, possibly correlated pleiotropic effects. In particular, median- and mode-based ratio estimators have been proposed for removing effects of invalid instruments under different assumptions[20,21]. Further, a recent study proposed the use of methods for outlier detection for conducting sensitivity analysis to the presence of invalid instruments[22]. While these and other new methods present important progress, there remain important gaps as they can be susceptible to substantial residual bias in the presence of a large number of invalid instruments or/and can produce estimates of causal effects with large uncertainty.

In this article, we propose a novel method for estimation of causal effects in multi-marker MR analysis by taking advantage of a working parametric model for the underlying bivariate effect-size distribution of the SNPs across pairs of traits. The model allows genetic correlation to arise both from causal and non-causal relationships. This model implies the zero modal pleiotropy (ZEMPA) assumption, which is also required by the mode-based estimator. For robust estimation of causal effects, we propose an estimating equation approach that essentially requires maximization of the probability concentration of the residuals, $\hat{\beta}_{\text{Trait2}} - \theta\hat{\beta}_{\text{Trait1}}$, at the null component of a normal-mixture model (method named MRMix, see Fig. 1 for an overview). We use extensive simulation studies to show that the proposed method can provide much better trade-off between bias and variance than existing estimators in a wide set of scenarios. We apply the proposed and existing methods for conducting MR analysis across a variety of exposures and health outcomes using publicly available summary-statistics from very large GWAS. The analysis reveals important differences across methods and new insights to causal relationships underlying some of these traits.

## Results

**Simulations.** Simulation studies show that MRMix can be far more robust compared to existing alternatives in a wide range of scenarios (Figs 2 and 3, Supplementary Figure 1). For example, when genetic correlation due to the causal relationship and pleiotropic effects are in the same direction (Fig. 2), MRMix generally produced nearly unbiased estimates of causal effects as long as the sample size for GWAS for the exposure (X) and the corresponding number of instruments reached a minimum threshold (e.g., $n_x > 100\,K$, $K > 100$). The bias was minimal or moderate even when only 25% of the instruments were valid. Among the alternatives, the inverse-variance weighted (IVW) and

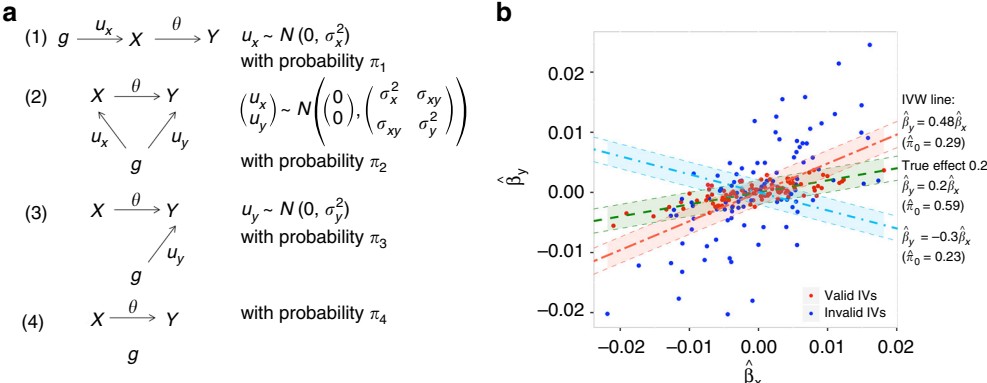

**Fig. 1** Overview of MRMix approach. **a** Four components of the Mendelian randomization mixture (MRMix) model: (1) Direct effect on X and an indirect effect on Y only through X; (2) direct effects on both X and Y; (3) direct effect on Y but no relationship with X; (4) related to neither X nor Y. Direct effects are denoted by $u_x$ and $u_y$ and the causal effect is denoted by $\theta$. Parameters $\pi_1$, $\pi_2$, $\pi_3$, $\pi_4$ are the mixing probabilities associated with components (1), (2), (3), and (4), respectively; within each component, we assume $u_x$ and $u_y$ to either be 0 or follow normal distributions and $\sigma_x^2, \sigma_y^2, \sigma_{xy}$ are the variance-covariance parameters. **b** Schematic of the MRMix estimation algorithm. The line corresponding to the true causal effect (0.2) maximizes the number of points "close" to the line. Here the regions that are covered by the 95% probability band under the null distribution (shaded area, expressed as $\left|\hat{\beta}_y - \theta\hat{\beta}_x\right| \leq 1.96\sqrt{s_y^2 + \theta^2 s_x^2}$) are highlighted. We plot the line $\hat{\beta}_y = \theta\hat{\beta}_x$ corresponding to three values $\theta = -0.3, 0.2, 0.48$; $\hat{\pi}_0$ is the estimated proportion of valid IVs. Source data are provided as a Source Data file

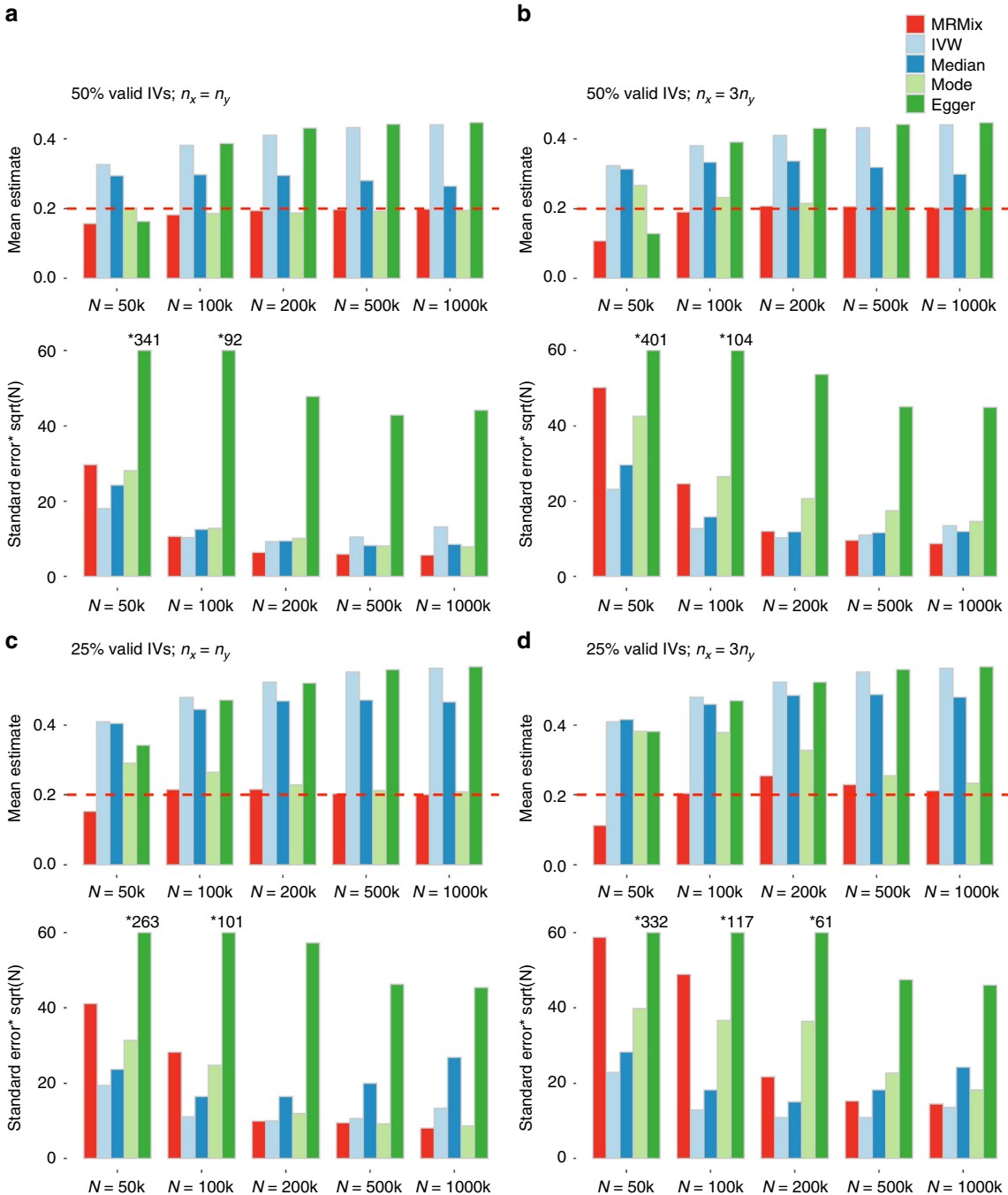

**Fig. 2** Simulation results when genetic correlation due to causal and pleiotropic effects are in the same direction. Mean and standard deviation of causal estimates are reported over 100 simulations. The true causal effect $\theta$ is 0.2. Estimates of association coefficient for SNPs across two traits are simulated assuming an underlying four-component model for effect-size distribution (Scenario A, see Methods), where SNPs could have direct effects on neither traits, only on $X$, only on $Y$, or on both with the effects being correlated. The proportion of valid instruments, i.e., the SNPs which have only direct effects on $X$, as a proportion of the total number of SNPs which are associated with $X$, are fixed at 50% or 25%. $n_x = N$: sample size of the study associated with $X$; $n_y$: sample size of the study associated with $Y$. Standard error bars higher than 60 are truncated and marked with *true-value. The average number of IVs, defined as the SNPs which reach genome-wide significance (z-test $p < 5 \times 10^{-8}$) in the study associated with $X$, is 14, 105, 399, 1135, and 1780 for $N = 50$ k, 100 k, 200 k, 500 k, and 1000 k, respectively. Source data are provided as a Source Data file

the Egger regression methods had the largest bias in directions away from null; the weighted median method was less sensitive, but had a considerable bias in many scenarios, and the weighted mode method had least bias. The bias of the weighted mode method was comparable to that of MRMix in most scenarios when the number of valid instruments was 50%, but was substantially more when the number of valid instruments dropped to

25% and the sample size for the GWAS of the outcome ($Y$) was relatively small (e.g., $n_x = 100$ K, $n_y = 33.3$ K).

When the genetic correlations due to causal relationship and pleiotropic effects were in the opposite directions (Fig. 3) MRMix showed more notable bias in estimation of causal effect—the direction of bias was generally towards the null and did not lead to estimates that were in the opposite direction of the true effect.

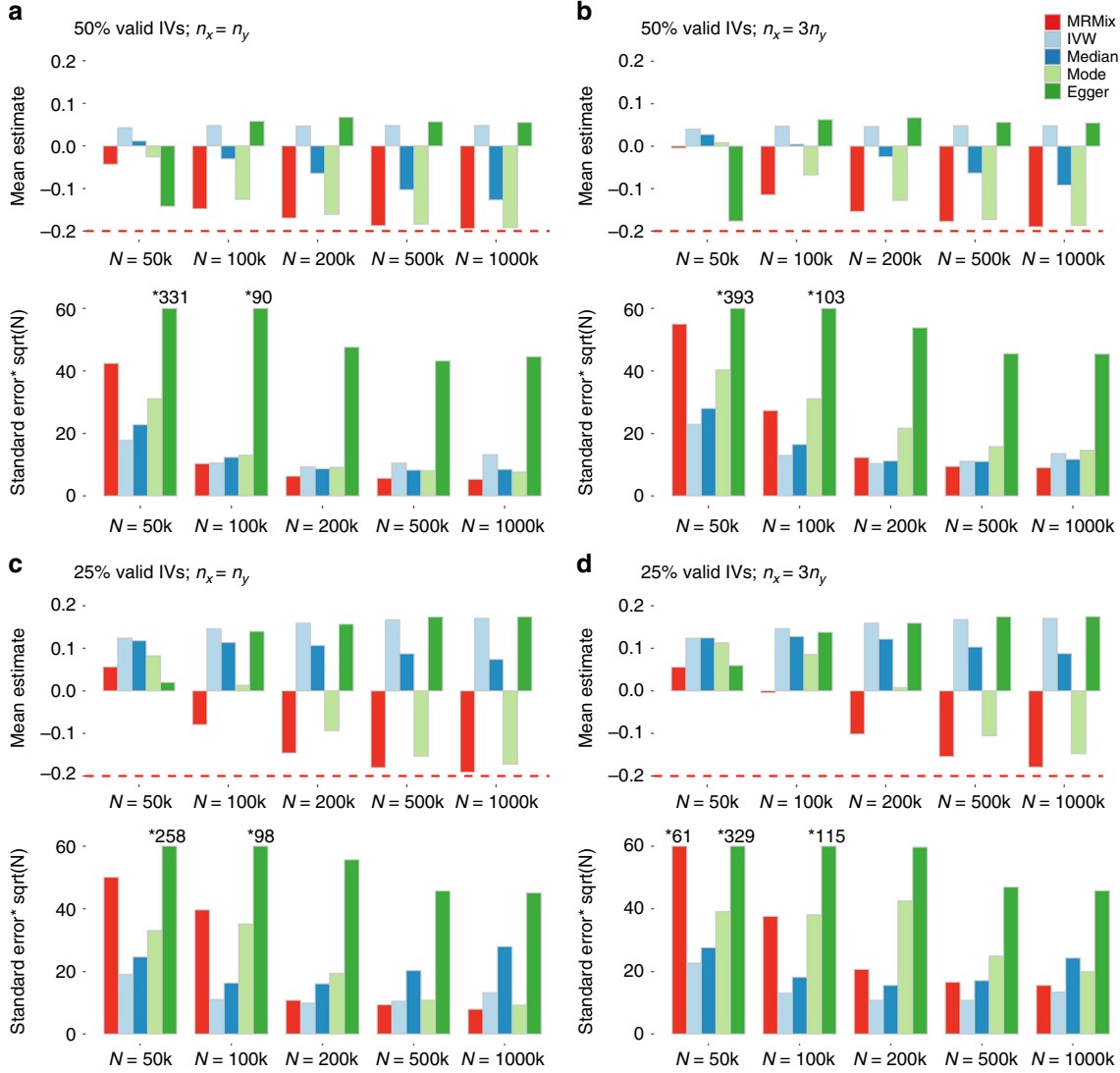

**Fig. 3** Simulation results when genetic correlation due to causal and pleiotropic effects are in opposite directions. Mean and standard deviation of causal estimates are reported over 100 simulations. The true causal effect $\theta = -0.2$. Estimates of association coefficient for SNPs across two traits are simulated assuming an underlying four-component model for effect-size distribution (Scenario A, see Methods), where SNPs could have direct effects on neither traits, only on $X$, only on $Y$, or on both with the effects being correlated. The proportion of valid instruments, i.e. the SNPs which have only direct effects on $X$ as a proportion of the total number of SNPs which are associated with $X$, are fixed at 50% or 25%. $n_x = N$: sample size of the study associated with $X$; $n_y$: sample size of the study associated with $Y$. Standard error bars higher than 60 are truncated and marked with *true-value. The average number of IVs, defined as the SNPs which reach genome-wide significance (z-test $p < 5 \times 10^{-8}$) in the study associated with $X$, is 14, 105, 399, 1135, and 1780 for $N = 50$ k, 100 k, 200 k, 500 k and 1000 k, respectively. Source data are provided as a Source Data file

The degree of bias was more when the number of valid instruments was smaller, but the bias steadily disappeared with increasing sample size irrespective of the proportion of valid instruments. In this scenario, the bias of all the other methods were much more severe and sometimes led to average estimates of causal effects in the opposite direction as those of the true effects. As earlier, the weighted mode method was the most robust among the alternatives considered, and yet it produced substantially more biased estimate of causal effect compared to MRMix in a number of scenarios.

Simulation studies also reveal MRMix estimates have much higher precision, i.e., smaller standard errors, relative to comparably robust estimators. In particular, the relative efficiency of MRMix compared to the weighted mode estimator, evaluated as the inverse of the ratio of respective variances, reached up to 3–4 fold in some of the settings. As expected, the IVW method generally had the smallest standard errors across all the scenarios,

but because it produced severe bias its efficiency is not comparable to that of MRMix. There were also several scenarios where the weighted median estimator had smaller standard errors compared to MRMix, but in all of these cases the former method produced substantially more biased estimates. Finally, across all scenarios, the Egger regression method produced estimates with much larger standard errors than the alternatives.

Reverse directional MR analysis shows that MRMix is highly sensitive to the causal direction (Supplementary Figures 2, 3). In contrast, the IVW, Egger regression and the weighted median method often produced estimates of causal effect of substantial magnitude from the outcome ($Y$) to the exposure ($X$). The weighted mode method, similar to MRMix, was found to be robust in the regard. In alternative simulation scenario, where we allow SNPs with larger effects to be more likely to be valid IVs, the methods rank similarly as described above, although all the methods tend to be less biased (Supplementary Figures 4, 5).

When the effect-sizes are generated from non-normal distributions, MRMix shows similar robustness and efficiency gain compared to MR-mode (Supplementary Figures 6–9). Simulation studies also showed that when the number of selected instruments were large, the analytical formula of standard error through asymptotic theory is generally quite accurate (Supplementary Table 1). When $n_x$ is between 100 K and 200 K, the estimator is conservative in the sense that it leads to some degree of overestimation of the true standard error.

**Data analysis**. We summarized the datasets we used in Supplementary Table 2. The MRMix analysis detected significant causal effects of genetically determined LDL-C, BMI, and blood pressure, but not that for HDL-C and triglycerides (TG), on the risk of coronary artery diseases (CAD) (Table 1). There were important differences across the methods in estimates of the causal effect for some of these factors. In particular, both IVW and the weighted median method detected significant causal effects of HDL-C and triglycerides, in directions consistent with known epidemiologic associations. The weighted mode method detected some effect for triglycerides, but the estimate had large standard error and was not statistically significant. The MRMix method estimated the causal effect for both of these lipid factors virtually to be zero. All methods detected causal effect of LDL-C in the expected direction and produced estimate of effect-size in similar range with respect to each other (OR for CAD per SD unit increase in LDL ranged between 1.28 and 1.51), but notably lower than those reported by previous MR analysis based on smaller number of genetic instruments[23]. Almost all methods detected causal effect of blood pressure and BMI in directions consistent with epidemiologic studies and produced estimates of effect-size in similar range. Egger regression and MR-mode yielded substantially wider confidence intervals thus leading to statistically non-significant or borderline significant results.

The MRMix analysis detected the significant causal effect of genetically determined BMI on the risk of breast cancer (BC). The method also detected suggestive evidence for causal effects for HDL-C and age-at-menarche (AAM), but not those for height, LDL-C, and TG. There were, again, important differences across methods. MRMix inferred negative causal relationship between increased level of BMI and the risk of BC, inconsistent with positive association that is typically seen in epidemiologic studies.

A previous MR analysis[24] that used fewer genetic instruments also detected the negative direction of the causal effect, but they reported the estimated effect-sizes to be somewhat stronger (OR for per SD unit increase in BMI reported to be in the range 0.56–0.75 compared to 0.84 by MRMix in the current study). The estimates from all the other methods were also in the negative direction, but those obtained from the weighted mode and Egger regression did not reach statistical significance due to large confidence intervals.

MRMix method indicated an increased level of HDL-C could be causally related to higher risk of BC (OR = 1.27 per SD increase in HDL-C level), but the result was only borderline statistically significant. The IVW and weighted median methods also detected these effects in the same direction, but the estimated effect-sizes were notably (by 50%) smaller. The weighted mode and Egger regression methods did not detect the effect to be statistically significant due to large confidence intervals. The MRMix was the only method which detected suggestive evidence of the casual effect of AAM on the risk of BC and the direction of effect was consistent with known epidemiologic association. Intriguingly, one previous study also noted that the standard IVW analysis does not detect any causal effect of AAM on the risk of breast cancer[25]. However, significant causal effect, in the same direction as the MRMix, has been reported in previous MR analysis which had adjusted for genetic relationship between AAM with BMI[25,26]. Finally, none of the methods detected a significant causal relationship between height and risk of BC although epidemiologic studies have consistently reported a positive association.

MRMix detected that genetically determined BMI increases the risk of major depressive disorder (MDD). All the other methods detected the same directional effect, but the magnitude of effect-sizes were notably smaller for the IVW, weighted median and Egger regression compared to the weighted mode and the MRMix method. The MRMix estimate for the effect of genetically determined years of education (EDY) on MDD had very large confidence interval and indicated no evidence of statistical significance for the causal effect. In contrast, the IVW and the weighted median methods detected a statistically inverse causal relationship between EDY and MDD. The weighted mode method also estimated the effect to be in the same direction, but the magnitude of the effect was attenuated and did not reach statistical significance.

**Table 1 Estimates and 95% confidence intervals for causal effects (log-OR of disease per SD-unit increase in risk-factor) of various putative risk-factors on three disease outcomes**

| Disease[a] | Risk-factors[b] | # of IVs[c] | MRMix | IVW | Weighted median | Weighted mode | Egger | LDSC[d] |
|---|---|---|---|---|---|---|---|---|
| CAD | BMI | 972 | 0.39 [0.32, 0.46] | 0.34 [0.3, 0.39] | 0.36 [0.3, 0.42] | 0.23 [−0.02, 0.48] | 0.76 [0.44, 1.08] | 0.44 |
| | LDL | 155 | 0.33 [0.23, 0.43] | 0.28 [0.21, 0.35] | 0.28 [0.22, 0.35] | 0.25 [0.04, 0.45] | 0.41 [0.05, 0.76] | 0.14 |
| | HDL | 200 | −0.01 [−0.12, 0.1] | −0.17 [−0.24, −0.1] | −0.08 [−0.15, −0.02] | −0.01 [−0.2, 0.17] | 0.23 [−0.09, 0.54] | −0.28 |
| | TG | 128 | −0.04 [−0.25, 0.17] | 0.24 [0.16, 0.31] | 0.19 [0.11, 0.28] | 0.14 [−0.14, 0.42] | 0.23 [−0.12, 0.58] | 0.26 |
| | SBP | 215 | 0.49 [0.33, 0.65] | 0.44 [0.34, 0.54] | 0.44 [0.35, 0.54] | 0.43 [0.17, 0.69] | 0.5 [−0.18, 1.18] | 0.54 |
| | DBP | 237 | 0.4 [0.24, 0.56] | 0.4 [0.31, 0.5] | 0.4 [0.31, 0.5] | 0.36 [0.28, 0.45] | 0.3 [−0.37, 0.97] | 0.58 |
| BC | BMI | 839 | −0.18 [−0.28, −0.08] | −0.1 [−0.16, −0.05] | −0.14 [−0.2, −0.08] | −0.15 [−0.45, 0.14] | −0.14 [−0.5, 0.21] | −0.19 |
| | Height | 3794 | 0.03 [−0.02, 0.08] | 0.02 [0, 0.04] | 0.02 [0, 0.05] | 0.01 [−0.15, 0.17] | 0.04 [−0.11, 0.18] | 0.08 |
| | LDL | 125 | 0.11 [−0.24, 0.46] | 0.06 [−0.01, 0.12] | 0.08 [0.01, 0.15] | 0.09 [−0.11, 0.3] | 0.12 [−0.2, 0.45] | 0.02 |
| | HDL | 152 | 0.24 [0, 0.48] | 0.09 [0.04, 0.15] | 0.11 [0.05, 0.17] | 0.13 [0, 0.27] | 0.03 [−0.22, 0.27] | 0.1 |
| | TG | 104 | −0.07 [−0.43, 0.29] | −0.06 [−0.13, 0.01] | −0.07 [−0.15, 0] | −0.11 [−0.25, 0.03] | −0.2 [−0.52, 0.11] | −0.04 |
| | Age at menarche | 262 | −0.13 [−0.28, 0.02] | −0.02 [−0.08, 0.04] | −0.05 [−0.11, 0.02] | −0.08 [−0.25, 0.1] | 0.18 [−0.24, 0.59] | 0.21 |
| MDD | BMI | 971 | 0.34 [0.09, 0.59] | 0.14 [0.09, 0.2] | 0.19 [0.12, 0.25] | 0.39 [0.01, 0.76] | 0.11 [−0.24, 0.45] | 0.14 |
| | Years of education | 510 | −0.18 [−2.18, 1.82] | −0.31 [−0.39, −0.23] | −0.3 [−0.39, −0.2] | −0.23 [−0.57, 0.11] | −0.4 [−0.97, 0.18] | −0.52 |

[a]CAD: coronary artery disease; BC: breast cancer; MDD: major depressive disorder
[b]LDL: low-density lipoprotein cholesterol. HDL: high-density lipoprotein cholesterol. TG: triglycerides. DBP: diastolic blood pressure. SBP: systolic blood pressure
[c]IVs are defined as SNPs which reach genome-wide significance (z-test $p < 5 \times 10^{-8}$) in the study associated with X
[d]LDSC: LD score regression estimates of causal effects is defined as $\rho_g / h_x^2$, the ratio between the estimated genetic covariance and the estimated heritability of the exposure (see Supplementary Notes for details)

## Discussion

In this article, we develop a novel and powerful method for conducting MR analysis using a large number of genetic instruments based on normal-mixture models for effect-size distribution where distinct mixture components are incorporated to allow genetic correlations to arise both from causal and non-causal relationships. To gain robustness against possible model mis-specification, we do not directly rely on the likelihood for model-based inference. Instead, we develop an estimating equation approach, that, in essence, involves estimation of causal effect through maximization of the probability concentration of residuals—defined by the total effect of SNPs on one trait after subtracting off indirect effects through the other trait—at the null component of a two-component normal mixture model. Both simulation studies and extensive data analyses show the method is not only robust, i.e., immune to bias in the presence of a large number of invalid instruments, but also can be highly efficient, i.e., it produces substantially more precise estimates of putative causal effects compared to alternative robust methods. The investigations also show the method is sensitive to the direction of causality and hence suitable for bi-directional MR analysis.

Simulation studies clearly demonstrate the superior performance of MRMix compared to a number of existing popularly used methods for MR analysis (Figs 2 and 3, Supplementary Figures 1–9). Stability of the method does require an adequately large sample size for the GWAS of the putative exposure of interest so that the number of instrument available for the analysis is reasonably large (e.g., >50). Once such threshold is exceeded, the method appears to be highly adaptive in dealing with invalid instruments and can maintain excellent trade-off between bias and efficiency compared to other methods. Even in the presence of large number of invalid instruments, the method often produces unbiased estimates of causal effect and, in settings, where there was notable bias, the bias was generally towards null and disappeared with increasing sample size. In the same settings, the alternative methods generally produced much larger bias, sometimes in the directions away from null and the bias always does not diminish with sample size. Among the alternatives considered, the mode-based ratio estimator shows similar level of robustness as MRMix for large sample size, which is intuitive given that both MRMix and mode-base estimator relies on the ZEMPA assumption. In spite of this similarity, for smaller sample size, in several settings, MRMix produces distinctly smaller bias. Further, MRMix clearly produces estimates with much smaller standard errors and this gain in efficiency is more pronounced when the number of valid instruments is larger, demonstrating the ability of the method to more effectively use the valid instruments compared to the weighted mode estimator. Although tuning the bandwidth of the mode-base estimator could improve its stability and efficiency, it is an additional difficulty for the users to deal with.

The MR analysis of the causal relationship of age-at-menarche and the risk of breast cancer provides an important empirical illustration of the strength of MRMix. It has been previously reported that genetic correlations between AAM and BC—due to underlying direct causal effect and that due to confounding/mediating effect of BMI—acts in opposite directions. Thus, polygenic MR analysis using standard IVW method could fail to identify the causal relationship[25]. However, when the IVW estimator is adjusted for the relationship of AAM associated SNPs with BMI, evidence of the casual effect of the inverse relationship between AAM and BC risk, consistent with epidemiologic observation, has been reported[26]. Consistent with previous studies, in our analysis, the standard IVW method produced estimate of causal effect for AAM to be virtual null. The MRMix, although did not explicitly account for BMI, produced estimate of

causal effect that is similar as those reported from BMI adjusted IVW in previous studies[25,26]. The weighted mode method, while pulled the estimate more towards the right direction compared to IVW, the estimate was attenuated and had very wide confidence intervals. Further, bi-directional MR analysis between BMI and AAM using MRMix suggested that genetically predicted BMI has an inverse causal effect on AAM, and the reverse directional effect is much weaker (Supplementary Tables 3, 4), and thus it appears that the SNPs which are associated with AAM through BMI, which, on its own, influences the risk of BC, are the underlying invalid instruments. The example demonstrates that MRMix has the ability to produce robust and efficient estimate of causal effects in the presence of potentially unobservable confounding factors.

Additional data analyses also illustrated the distinct property of MRMix compared to alternatives. In particular, the MRMix method estimated the causal effect of HDL and triglycerides on the risk of CAD to be virtually null, while standard IVW and weighted median methods detected these effects to be significantly away from null in directions consistent with known epidemiologic associations. A recent study reported the significant putative causal effect of years of education on reducing risk of major depressive disorder based on standard IVW analyis[27]. MRMix analysis produced large degree of uncertainty in the underlying estimate of causal effect and did not provide any evidence of statistical significance. MRMix found the causal effect of genetically determined BMI on increasing the risk of MDD to be notably stronger than that is indicated by the traditional IVW method. Thus, it is possible that there are common genetic pathways underlying these traits, which lead to genetic correlation in opposite directions than that is due to the direct causal effect of BMI on MDD.

Recently a number of alternative methods have been proposed for conducting robust MR analysis in the presence of invalid instruments. One such method, termed as MR-PRESSO[16], applies outlier detection test to each individual genetic variant and removes potentially invalid instruments. While the method was shown to be highly useful for the detection of bias in reported estimates of causal effects in existing MR analysis, the method can only partially correct for bias and relies on the InSIDE assumption. Further, because the method requires conducting a series of tests and evaluating their significance based on simulations, implementation of it can be time-consuming and estimation of uncertainty associated with the final estimator can be challenging. A related method for instrument selection, termed two stage hard thresholding (TSHT) with voting, constructs many estimates of the set of valid IVs and use majority or plurality voting to make final decisions[28]. This method was proved to be consistent in instrument selection and effect estimation, but requires individual-level data and is not as widely applicable as summary level data based methods. Another method proposes to obtain IVW estimators for all possible subsets of genetic instruments and then combine them with a model averaging method with lower weight given to more heterogeneous subsets[29]. While the method is shown to be highly robust as well as powerful, it is currently not scalable for the analysis of large number of instruments which is the focus of the current study.

Another study proposed analysis of the causal relationship between traits based on genetic relationship, but using a different framework than that for the standard MR analysis. The study defined one trait to be partially or fully causal for another, if there is an underlying genetically determined latent variable which influences both traits, but has a stronger relationship with the first than the second[30]. The study defined moment equations for the estimation of parameters quantifying degree of partial causality using GWAS summary-statistics. We believe this novel

framework and more traditional MR hypothesis can complement each other to provide an improved understanding of the nature of genetic correlation across traits. The use of latent variable framework, for example, detected evidence of partial causality of several cholesterol traits on blood pressure level. Neither MRMix, nor any of the other methods, detected any evidence of direct causal effects underlying these traits (see Supplementary Table 3). Thus the evidence of partial causality is likely to have been primarily driven by the existence of underlying common genetic pathways which are more strongly related to cholesterol level than blood pressure.

The MRMix method has limitations as well. First, the method relies on certain model for underlying effect-size distribution. As we have noted before, the mis-specification of effect-size distribution is not as critical as we do not use the model directly to perform maximum-likelihood estimation, but instead use the model as an efficient way of identifying certain "mode"-based estimator based on underlying estimating equations. Simulation studies show that even when the underlying effect-size distribution has more complexity than the assumed model, the estimation of causal effect parameters can remain relatively robust. Nevertheless, more extensive simulation and theoretical studies are needed to further understand the property of the method under complex but realistic models for effect-size distributions as has been evidenced from recent study[5]. Second, the method does require pre-selection of SNPs as genetic instruments based on p-value in the z-test for the significance of their association. We have observed that as long as the significance threshold is stringent (e.g., p-value $<5 \times 10^{-8}$), there is not substantial winner's curse bias due to the selection of SNPs and estimation of their coefficients from the same study (Supplementary Table 5). While the method, in principle, can be extended to include SNPs with more liberal threshold, it can suffer from winner's curse bias unless the SNP selection and coefficient estimation are performed based on independent studies (Supplementary Table 5). Third, in our current study, we have focused on the analysis of independent SNPs selected from GWAS through stringent LD-pruning after prioritizing by p-values. As we perform MR analysis based on the marginal effects of the individual SNPs, some of which may tag multiple underlying causal SNPs, the underlying pattern of LD may cause some bias in MRMix as well the other methods. Further studies are needed to investigate the effect of LD in MR analysis, especially when large number of genetic instruments are used. Fourth, though our simulation settings are flexible and realistic, they assume zero modal pleiotropy. Further studies are needed to investigate the performance in scenarios where ZEMPA is violated or scenarios that favor other methods (e.g., median or Egger regression). Further studies are also merited to explore the property of MRMix under more complex causal structure in the data, such as partial causality[30] and multiple components of causality[29].

In conclusion, MRMix provides a novel tool for conducting robust and powerful MR analysis using large number of genetic instruments that are now rapidly becoming available from the recent expansion of GWAS. We demonstrate through simulation studies, as well as variety of real data analyses, that the method has notable ability to trade-off bias and efficiency for estimation of causal effects in the presence of invalid instruments. Application of MRMix for future MR studies will lead to improved understanding of causal basis of genetic correlation across traits.

## Methods

**Model setup.** We propose a method for two-sample MR analysis that requires only summary-level GWAS association statistics for a putative exposure ($X$) and the outcome ($Y$) from separate studies. We describe the proposed method in the context of independent SNPs. Let $(\beta_{jx}, \beta_{jy})$, $j = 1,2,\ldots M$, denote the underlying true association coefficients in a standardized scale for the $M$ SNPs for the exposure ($X$) and the outcome ($Y$). The standard MR analysis assumes that all the SNPs are valid instruments, i.e., they are associated with $X$ but have no direct effect on $Y$. If the assumption is satisfied, then the two sets of regression coefficients will satisfy a proportional relationship in the form $\beta_y = \theta\beta_x$, where $\theta$ is the causal effect of $X$ on $Y$.

The proportional relationship holds if $X$ and $Y$ follow linear models of the form $Y = \alpha_y + \theta X + \epsilon_y$ and $X = \alpha_x + \beta_x G + \epsilon_x$ with the assumption that $E(\epsilon_y | G, X) = E(\epsilon_y | X) = 0$ and $E(\epsilon_x | G) = 0$. Further, the relationship holds for binary $Y$ if $X$ and $Y$ follow log-linear model $P(Y = 1|X) = \exp(\alpha_y + \theta X)$ and the regression model for $X$ is $X = \alpha_x + \beta_x G + \epsilon_x$, where $\epsilon_x$ is independently distributed of $G$. Then we can derive $P(Y = 1|G) = \exp(\alpha_y)E\{\exp(\theta X)|G\} \propto \exp\{\theta\beta_x G\}$. The proportionality assumption also approximately holds under logistic regression model when the outcome is relatively rare (in the population) under which log-linear and logistic models become similar. In our data analysis for disease outcomes, we assume that the proportionate relationship holds in the log-odds-ratio parameter scale[31].

Instead of assuming the proportional relationship holds across all instruments, we propose modeling the bivariate effect-size distribution using a flexible normal-mixture model where the proportional relationship needs to be satisfied only for a fraction of the genetic variants.

We assume a SNP can have four different types of effects: (1) direct effect on $X$ and an indirect effect on $Y$ only through $X$, (2) direct effects on both $X$ and $Y$, (3) direct effect on $Y$ but no relationship with $X$, (4) related to neither $X$ nor $Y$ (Fig. 1a). If we let $u_x$ and $u_y$ be the direct effects of a SNP on $X$ and $Y$, respectively, then we can write $\beta_x = u_x$ and $\beta_y = u_y + \theta u_x$. We also make distributional assumptions for the four types of effects: (1) $u_x \sim N(0, \sigma_x^2)$, $u_y = 0$, (2)

$$\begin{pmatrix} u_x \\ u_y \end{pmatrix} \sim N\left( \begin{pmatrix} 0 \\ 0 \end{pmatrix}, \begin{pmatrix} \sigma_x^2 & \sigma_{xy} \\ \sigma_{xy} & \sigma_y^2 \end{pmatrix} \right),$$ (3) $u_x = 0, u_y \sim N(0, \sigma_y^2)$, (4) $u_x = u_y = 0$.

The first component includes the valid IVs. The second component includes SNPs in horizontal pleiotropy, i.e., SNPs with potentially correlated effects on $X$ and $Y$. This component allows violation of the InSIDE assumption as we allow $\sigma_{xy} \neq 0$. The SNPs in third and fourth components are not associated with $X$, but can be included when we apply a liberal instrument selection threshold.

Note that our model implies the zero modal pleiotropy (ZEMPA) assumption, which is also required by the mode-based estimator[21]. To see this, note that the pleiotropic effect $u_y = 0$ in the first and fourth components, and follows continuous distribution $N(0, \sigma_y^2)$ in the second and third components. Hence the most common value of $u_y$ is 0, which is equivalent to the ZEMPA assumption.

**MR analysis using mixture-model (MRMix).** In GWAS, we obtain noised estimates $\hat{\beta}_x$ and $\hat{\beta}_y$, where one can assume $\hat{\beta}_x \sim N(\beta_x, s_x^2), \hat{\beta}_y \sim N(\beta_y, s_y^2)$, with known standard errors $s_x^2$ and $s_y^2$. In principle, a likelihood for the observed data can be written by integrating over the "prior" model for the bivariate effect-size distribution. However, maximum-likelihood estimation of the target parameter $\theta$, jointly with all of the nuisance parameters $(\pi_1, \pi_2, \pi_3, \sigma_x^2, \sigma_y^2, \sigma_{xy})$ may face computational challenges due to identifiability issues associated with mixture likelihoods. Further, the inference can be sensitive to violation of the underlying modeling assumptions.

In the following, we propose an alternative estimation procedure that is computationally simple and rely less on the underlying model assumption. This procedure effectively solves a spike-detection problem. Intuitively, we observe that under this model, the true causal effect $\theta$ maximizes the number of points "close" to the line $\hat{\beta}_y = \theta\hat{\beta}_x$, i.e. points for which the vertical distance $\hat{\beta}_y - \theta\hat{\beta}_x$ can be covered by the null distribution $N(0, s_y^2 + \theta^2 s_x^2)$. For a different value $\tilde{\theta}$ the null distribution $N(0, s_y^2 + \tilde{\theta}^2 s_x^2)$ covers a smaller number of points (Fig. 1b). We characterize this observation statistically as follows. The distribution of the residuals $\hat{\beta}_y - \theta\hat{\beta}_x$ can be written at the true ($\theta$) and alternative value ($\tilde{\theta}$) under the proposed model in the form

$$\hat{\beta}_y - \theta\hat{\beta}_x \sim (\pi_1 + \pi_4)N\left(0, s_y^2 + \theta^2 s_x^2\right) + (\pi_2 + \pi_3)N(0, \sigma_y^2 + s_y^2 + \theta^2 s_x^2)$$

$$\hat{\beta}_y - \tilde{\theta}\hat{\beta}_x \sim \pi_1 N\left(0, \left(\theta - \tilde{\theta}\right)^2 \sigma_x^2 + s_y^2 + \tilde{\theta}^2 s_x^2\right)$$

$$+ \pi_2 N\left(0, \sigma_y^2 + 2\left(\theta - \tilde{\theta}\right)\sigma_{xy} + \left(\theta - \tilde{\theta}\right)^2 \sigma_x^2 + s_y^2 + \tilde{\theta}^2 s_x^2\right)$$

$$+ \pi_3 N\left(0, \sigma_y^2 + s_y^2 + \tilde{\theta}^2 s_x^2\right) + \pi_4 N\left(0, s_y^2 + \tilde{\theta}^2 s_x^2\right)$$

See Supplementary Notes for details. Note that when $\tilde{\theta} = \theta$, the first and fourth terms collapse, leading to an enrichment of the point mass at $N\left(0, s_y^2 + \tilde{\theta}^2 s_x^2\right)$. Only at the true value $\theta$, $\hat{\beta}_y - \theta\hat{\beta}_x$ does have an enriched point mass $\pi_1 + \pi_4$ at $N\left(0, s_y^2 + \theta^2 s_x^2\right)$, while for other values $\tilde{\theta}$ this point mass is $\pi_4$. The enrichment $\pi_1$

is contributed by the SNPs that have no direct effects on $Y$, the key assumption underlying instrumental variable (IV) method. Our approach uses this property to identify the causal effect.

Based on the above observations, we propose the following estimation procedure:

(i) For a fixed $\bar{\theta}$, perform maximum-likelihood to fit the two-component normal mixture model in the form $\hat{\beta}_y - \bar{\theta}\hat{\beta}_x \sim \pi_0 N\left(0, s_y^2 + \bar{\theta}^2 s_x^2\right) + (1 - \pi_0)N(0, \sigma^2)$ to get estimates of unknown parameters as $\hat{\pi}_0(\bar{\theta})$ and $\hat{\sigma}^2(\bar{\theta})$;

(ii) Search over a grid of $\bar{\theta}$ values and choose the one that maximizes $\hat{\pi}_0(\bar{\theta})$ as the estimate, i.e., $\hat{\theta} = argmax_{\bar{\theta}}[\hat{\pi}_0(\bar{\theta})]$.

Under the working model $\hat{\beta}_y - \bar{\theta}\hat{\beta}_x \sim \pi_0 N\left(0, s_y^2 + \bar{\theta}^2 s_x^2\right) + (1 - \pi_0)N(0, \sigma^2)$, $\pi_0$ is the proportion of valid IVs and $\sigma^2$ is the unknown variance parameter associated with the invalid IVs. We note that in step (i), for computational simplification, we are only fitting a two-component normal mixture model which is correct when $\bar{\theta} = \theta$ (the true value). When $\bar{\theta} \neq \theta$, the two-component model is not correct and can only provide an approximation of the underlying multi-component normal mixture model. We observe in simulation studies that although the model is wrong under the alternative, the proposed estimate $\hat{\theta}$ has no asymptotic bias. In contrast, a maximum-likelihood estimator, which maximizes the likelihood of the residuals under the two-component normal-mixture model, produces substantially biased estimate of causal effect due to mis-specification of the model under alternative.

If the study for $X$ and $Y$ have overlapping subjects, the null component $N\left(0, s_y^2 + \bar{\theta}^2 s_x^2\right)$ in step (i) can be easily modified to account for correlation in estimated effects. For example, one could use the bivariate LD score regression[13,32] to estimate, $s_{xy} = cov(\hat{\beta}_x, \hat{\beta}_y | \beta_x, \beta_y)$, the covariance of GWAS estimates given the effect-sizes. Hence the null component could be modified as $N\left(0, s_y^2 + \bar{\theta}^2 s_x^2 - 2\bar{\theta}s_{xy}\right)$ to account for sample overlap across studies of $X$ and $Y$.

**Variance estimation.** In this section, we use asymptotic theory to derive the standard error of MRMix estimator. Although the inference for spike-detection is non-standard, we can derive an underlying estimating equation by exploiting the fact that $\theta$ maximizes $\hat{\pi}_0(\theta)$, which itself is obtained by maximization of a parametric likelihood. Note that the value of $\theta$ that maximizes $\hat{\pi}_0(\theta)$ can be found by solving equation $\partial\hat{\pi}_0/\partial\theta = 0$. We can express $\partial\hat{\pi}_0/\partial\theta$ in terms of the parametric likelihood using implicit function theorem. For each $\theta$, we fit a two-component normal mixture model with log-likelihood $l\left(\theta, \pi_0, \sigma^2 | \hat{\beta}_{1x}, \hat{\beta}_{1y}, \ldots, \hat{\beta}_{Mx}, \hat{\beta}_{My}\right)$ to estimate $\pi_0$ and $\sigma^2$. The score equations are $\partial l/\partial\pi_0 = 0, \partial l/\partial\sigma^2 = 0$. Using implicit function theorem, we can show that solving $\partial\hat{\pi}_0/\partial\theta = 0$ is equivalent to solving

$$\frac{\partial^2 l}{\partial(\sigma^2)^2}\frac{\partial^2 l}{\partial\pi_0\partial\theta} - \frac{\partial^2 l}{\partial\sigma^2\partial\pi_0}\frac{\partial^2 l}{\partial\sigma^2\partial\theta} = 0 \quad (1)$$

under conditions $\pi_0 = \hat{\pi}_0(\theta, \hat{\beta}_{1x}, \hat{\beta}_{1y}, \ldots \hat{\beta}_{Mx}, \hat{\beta}_{My})$ and $\sigma^2 = \hat{\sigma}^2\left(\theta, \hat{\beta}_{1x}, \hat{\beta}_{1y}, \ldots \hat{\beta}_{Mx}, \hat{\beta}_{My}\right)$. By taking the Taylor expansion of the estimating function with respect to first $\theta$ and then $\beta$'s, we obtained an influence function representation of the final estimator (See Supplementary Notes for details). The standard error of $\hat{\theta}$ can be easily calculated from the influence function representation.

**Simulation setup.** We conduct extensive simulation studies to evaluate the proposed method under different scenarios.

We simulate genome-wide summary statistics of 200,000 independent SNPs. We first simulate the direct effect-sizes $(u_x, u_y)$ and compute the total effect-sizes as: $\beta_x = u_x$ and $\beta_y = u_y + \theta u_x$. Then we generated the summary statistics by simulating independently from $\hat{\beta}_x \sim N\left(\beta_x, \frac{1}{n_x}\right)$ and $\hat{\beta}_y \sim N\left(\beta_y, \frac{1}{n_y}\right)$, where $n_x$ and $n_y$ are the sample sizes for studies associated with $X$ and $Y$, respectively. This mimics the two-sample MR setup where the exposure and the outcome are measured on independent samples. In all our simulations, we set $n_y = \frac{n_x}{C}$ where we vary to $C = 1, 3$ with the choice of $C = 3$ reflects scenarios where the effective sample size for $Y$ can be expected to be lower than the exposure $X$.

We simulated true effect-sizes for the SNPs under the hypothesized four-component model as well as more complex models that include additional mixture components. Under *Scenario A*, we simulated effect-sizes $u_x$ and $u_y$ from the four-component mixture model (Fig. 1a), where we vary the proportion of valid IVs by changing the ratio of $\pi_1$ and $\pi_2$ according to the following specifications:

50% causal SNPs for $X$ are valid IVs: $\pi_1 = \pi_2 = 0.01$.

25% causal SNPs for $X$ are valid IVs: $\pi_1 = 0.005, \pi_2 = 0.015$.

For both cases, we set $\pi_3 = 0.01, \pi_4 = 0.97, \sigma_x^2 = \sigma_y^2 = 5 \times 10^{-5}$ and $\sigma_{xy} = 0.5\sigma_x\sigma_y$. According to the model, a total of 2% of the SNPs have direct effect on $X$

and either 2% or 2.5% of the SNPs have direct effect on $Y$; and the overlapping SNPs have strongly correlated effects (correlation = 0.5). The total heritability of $X$ is 20% and that for $Y$ ranges from 18.8% to 28.8%, respectively. We allow the causal effect ($\theta$) of $X$ on $Y$ to be null, or non-null in positive or negative directions so that the genetic correlations due to pleiotropic and causal effects can act in opposite directions.

In *Scenario B*, we simulated effect-sizes using more complex normal mixture model that allows existence of clusters of non-null SNPs with distinctly larger effects than others[5]. In particular, we allow a fraction of causal SNPs for $X$ to have distinctly larger effects and we assume the SNPs that have larger effects are more likely to be valid IVs. Similarly, among SNPs which have direct effects on $Y$, we allow existence of SNPs which have distinctly larger effects and these SNPs are less likely to have direct effect on $X$. We allow SNPs with distinct cluster of effect-sizes through incorporation of additional normal-mixture components with varying variance-component parameters (see Supplementary Notes for more details).

Finally, in a third scenario (*Scenario C*), we conduct simulations to study the robustness of the proposed methods when effect-size distribution does not follow normal or mixture-normal forms. In particular, we simulated effect-sizes across $X$ and $Y$ using the same mixture model as depicted in Fig. 1a, but generated effect-sizes under each component using non-normal, but symmetric and unimodal distributions, such as Laplace and T distributions (Supplementary Notes).

**Inclusion of SNPs with liberal threshold.** In our main analysis, we focused on analysis based on SNPs as instruments that have achieved genome-wide significance in the study associated with $X$. We further explored the ability of the method to handle additional SNPs below genome-wide significance. When SNPs are included using more liberal threshold, one would expect a fraction of these SNPs to be null. In the presence of null SNPs, the probability concentration of the two-component mixture model for the residuals at the null component is $(\pi_1 + \pi_4)$, where $\pi_1$ is the proportion of valid instruments and $\pi_4$ is the proportion of null SNPs. Thus, while the inclusion of more SNPs as potential instruments could lead to increase in efficiency due to increase in the underlying valid instruments, if a very liberal threshold is used, then large value of $\pi_4$ can obscure estimation of $\pi_1$. Thus, one would expect there would be an optimal threshold for SNP selection, as is typically observed for building polygenic risk scores for risk-prediction. We varied the $p$-value threshold in the z-test for instrument selection to 0.005, $5 \times 10^{-4}$, $5 \times 10^{-6}$ and studied the bias and standard errors for resulting MRMix estimates. Further, as the winner's curse problem can create bias when selection of SNPs and estimation of their effects are done based on the same study, we also studied the performance of MRMix when effect-sizes of the SNPs associated with $X$ are estimated based on an independent dataset than the one used to select the SNPs.

**Summary level data.** We applied MRMix for the analysis of publicly available GWAS summary level data to explore causal relationships underlying a variety of exposure-outcome pairs of interest. We selected these pairs based on available sample sizes and number of underlying instruments, existing evidence of epidemiologic associations in the literature or/and evidence of causality from recent MR studies. On the exposure side, we accessed data for height and body mass index[4], blood lipids[33], education attainment[34], blood pressure[35] and age at menarche[26]. For data analysis, we only selected SNPs to be potential instruments if they reach genome-wide significance (z-test $p$-value $< 5 \times 10^{-8}$) in the respective studies. Further, we used LD-clumping with an $r^2$ threshold of 0.1 to select a set of independent instruments for each trait. The number of instruments across the different exposures varied between 104 and 3794, with the largest numbers being available for height ($K = 3794$), BMI ($K = 972$) and years of education ($K = 510$) due to the availability of results from the large UK Biobank study.

On the outcome side, we accessed data for coronary artery disease (CAD)[36] for its analysis in relationship to known major risk factors BMI, blood lipids and blood pressure[37–43]; breast cancer[44] in relationship to several known epidemiologic risk-factors, including height, age-at-menarche, BMI, cholesterol level[45–48]; and major depressive disorder (MDD)[27] in relationship to BMI and years of education[49–51]. In addition, we explored potential causal interrelationships among some of the exposures themselves, such as between BMI and blood pressure, and between BMI and age-at-menarche[52,53]. See Supplementary Table 2 for more information.

For all datasets, we only included SNPs among a set of ~1.07 million HapMap3 SNPs that have MAF > 0.05 and have matching alleles in the 1000 Genomes European sample. We set the first allele in 1000G data as effect allele, flipping the sign of the $\hat{\beta}$ coefficient when necessary. We also removed SNPs whose reported sample sizes were less than 2/3 of the 90th percentile of the sample size distribution across SNPs in respective studies. Finally, we removed SNPs in major histocompatibility complex (MHC) region (26 ~34 Mb on chromosome 6) and SNPs that have very large z score ($z^2 > 80$) to prevent the outliers that may unduly influence the results[54].

**Alternative methods.** For both simulations and real data applications, we compare MRMix with existing popularly used MR methods that allow the estimation of causal effects. In particular, we included inverse-variance weighted (IVW) method[10], weighted median[20], weighted mode[21] and Egger regression[17]. Further,

we observe that if the InSIDE assumption holds across all SNPs, then the LD score regression methodology[13] can be used to estimate the causal effect without any pre-selection of SNPs. In this case, the estimate is simply given by $\theta = \rho_g / h_x^2$, where $\rho_g$ is the estimated genetic covariance of the pair of traits and $h_x^2$ is the estimated heritability of X. This estimator is nearly equivalent to Egger regression using the same set of SNPs (see Supplementary Notes for details). In fact, any method to estimate heritability and genetic correlation can be used in this way to estimate causal effects. Thus, as a benchmark for comparison, in real data analysis, we also report estimates of causal effect based on the LD score regression.

**Reporting Summary**. Further information on experimental design is available in the Nature Research Reporting Summary linked to this article.

## Data Availability

The data used in this study are publicly available at the URLs below. 1000 Genomes Phase 3 European sample, HapMap3 SNP list, https://data.broadinstitute.org/alkesgroup/LDSCORE/; GIANT Consortium (BMI and height) summary statistics, http://portals.broadinstitute.org/collaboration/giant/index.php/GIANT_consortium_data_files; Global Lipids Genetics Consortium (cholesterol traits), http://csg.sph.umich.edu/abecasis/public/lipids2013/; Neal lab UK Biobank GWAS (blood pressure summary statistics), http://www.nealelab.is/blog/2017/7/19/rapid-gwas-of-thousands-of-phenotypes-for-337000-samples-in-the-uk-biobank; ReproGen Consortium (age at menarche), http://www.reprogen.org/data_download.html; Social Science Genetic Association Consortium (SSGAC), https://www.thessgac.org/data; CARDIoGRAMplusC4D Consortium (coronary artery disease), http://www.cardiogramplusc4d.org/data-downloads/; Breast Cancer Association Consortium (BCAC) summary statistics, http://bcac.ccge.medschl.cam.ac.uk/bcacdata/oncoarray/gwas-icogs-and-oncoarray-summary-results/; Psychiatric Genomics Consortium (PGC), https://www.med.unc.edu/pgc/results-and-downloads/downloads. The source data underlying Figs. 1b, 2, 3, Supplementary Figures 1–9 and Supplementary Tables 1 and 5 are provided as a Source Data file. All other relevant data are available upon request.

## Code availability

MRMix software is publicly available at: https://github.com/gqi/MRMix. The software has been tested on MAC OS 10.11.5 with R version 3.5.1. PLINK 1.9 was used to calculate linkage disequilibrium (LD) coefficients and hence perform LD clumping and calculate LD scores.

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

## Acknowledgements

The research was supported by funding from NIH for environmental of Child Health Outcomes (ECHO) Cohort Data Analysis Center (U24OD023382) and by the Bloomberg Distinguished Professorship Endowment at the Johns Hopkins University. Data on coronary artery disease have been contributed by CARDIoGRAMplusC4D investigators and have been downloaded from www.CARDIOGRAMPLUSC4D.ORG. The breast cancer genome-wide association analyses were supported by the Government of Canada through Genome Canada and the Canadian Institutes of Health Research, the 'Ministère de l'Économie, de la Science et de l'Innovation du Québec' through Genome Québec and grant PSR-SIIRI-701, The National Institutes of Health (U19 CA148065, X01HG007492), Cancer Research UK (C1287/A10118, C1287/A16563, C1287/A10710) and The European Union (HEALTH-F2-2009-223175 and H2020 633784 and 634935). All studies and funders are listed in Michailidou et al. (2017).

## Author contributions

G.Q. and N.C. conceived the methods and wrote the manuscript. G.Q performed all the data analysis and simulation studies.

## Additional information

**Competing interests:** The authors declare no competing interests.

