## [Peer Review File · Nature Communications]

Reviewers' comments:

Reviewer #1 (Remarks to the Author):

The authors present a new approach to estimating a causal effect by Mendelian randomisation. They allow SNP effects on exposure and outcome to follow a bivariate mixture distribution, and under a working model containing Normal and null components they estimate the causal effect as maximising the proportion of SNPs whose bivariate effects lie on the corresponding straight line through the origin. The method is shown to work well with large numbers of instrument SNPs, with similar levels of bias to the modal estimator of Hartwig et al but improved precision.

There is a lot to like about this paper, including a novel estimating equation approach to this problem, a computational simplification in the estimation procedure, and simulations that are more realistic than sometimes found in this literature.

Major comment:

1. The key assumption of this method is that the largest set of SNPs with a common θ , where $\beta_y = \theta \beta_x$, has θ equal to the causal effect. But the paper should be more clear on the central role of this assumption. At L153 it is introduced with "intuitively, we observe that...", but this is not an observation, it is an assumption, in fact the same as the ZEMPA assumption (zero modal pleiotropy) made in the modal estimator of Hartwig et al. It is thus unsurprising that the method has similar bias to Hartwig's approach. Only in the discussion L546 do the authors compare their method to a "mode"-base estimator. However, the authors' approach can readily be interpreted as estimating the mode of the distribution of ratio estimates. The advance of this paper is in the improved stability and robustness of the estimator.

I would suggest that the authors are clear from the start (ie introduction) that they are proposing a better (ie similar bias, better precision) estimator for the mode than the kernel density estimator of Hartwig et al, and that it similarly relies crucially on ZEMPA.

The authors might mention related work of Guo Z et al (JRSS(B) 2018, discussed in Hartwig et al) and Burgess et al (IJE 2018, <https://doi.org/10.1093/ije/dyy080>) who have another approach to mode estimation through mixture distributions.

Minor comments:

2. Of course, the simulations were also based on ZEMPA, and the present method would do worse under simulations that favoured MR-Egger or the weighted median. To include such simulations would lengthen the paper unnecessarily, but it would be worth acknowledging the fact.

3. L107, "we do not use the likelihood ... to reduce sensitivity" reads a bit clumsy, suggest rephrasing

4. L128 and throughout, make clear that we assume linear regression models for exposure and outcome - in fact it would be good to write the models out explicitly.

5. L233-4 "overlapping SNPs have strongly correlated effects" - what is the value of this correlation?

6. L267 typo, "there would be is an"

7. L313 the LD score regression method is I think novel in this paper, and nice. It is equivalent to Egger regression over the whole genome, as is clear from the derivation. It would be worth saying this explicitly, in addition to the current note that both methods assume InSIDE. In fact, any method that

estimates genetic correlation could be used in this way (GCTA, AVENGEME, ...)

8. L335 the performance of the mode estimator could probably be improved by adjusting the bandwidth; maybe state this as a limitation of that method.

9. L411 typo "indicted"

10. L475 type "casual"

11. L536 "as well as none of the other ... methods" - reads a bit clumsy, please rephrase

12. L572 please delete "unique" (other methods would perform better in situations that favored their assumptions)

Reviewer #2 (Remarks to the Author):

The authors could provide more details in motivating their mixture modeling methods. Typically the MA analysis starts with genetic variants from existing GWAS for the intermediate phenotypes, so that there is assured association between instrument and the intermediate phenotype. If so, why type (3) and (4) in Figure 1 is ever needed?? The authors need to start with the three classical IV assumptions including non-zero instrumental strength.

the notation in Methods is hard to follow: what is σ^2_x , σ^2_y and σ_{xy} ? define π_1 ... π_4 when they first appear. What is π_0 ? why (2) and (3) collapse in line 160. what is σ^2 in line 173? The derivation of methods in page need clarification/explanation.

The data section needs more description about sample size, where the data came from and what conclusion has been draw by previous analyses.

Response to reviewers

Reviewer #1 (Remarks to the Author):

General Comment

The authors present a new approach to estimating a causal effect by Mendelian randomisation. They allow SNP effects on exposure and outcome to follow a bivariate mixture distribution, and under a working model containing Normal and null components they estimate the causal effect as maximising the proportion of SNPs whose bivariate effects lie on the corresponding straight line through the origin. The method is shown to work well with large numbers of instrument SNPs, with similar levels of bias to the modal estimator of Hartwig et al but improved precision.

There is a lot to like about this paper, including a novel estimating equation approach to this problem, a computational simplification in the estimation procedure, and simulations that are more realistic than sometimes found in this literature.

Response

We would like to thank the reviewer for his/her encouraging comments. We found your comments to be very insightful and have addressed them in the revision.

Comment

The key assumption of this method is that the largest set of SNPs with a common θ , where $\beta_y = \theta \beta_x$, has θ equal to the causal effect. But the paper should be more clear on the central role of this assumption. At L153 it is introduced with "intuitively, we observe that...", but this is not an observation, it is an assumption, in fact the same as the ZEMPA assumption (zero modal pleiotropy) made in the modal estimator of Hartwig et al. It is thus unsurprising that the method has similar bias to Hartwig's approach. Only in the discussion L546 do the authors compare their method to a "mode"-base estimator. However, the authors' approach can readily be interpreted as estimating the mode of the distribution of ratio estimates. The advance of this paper is in the improved stability and robustness of the estimator.

I would suggest that the authors are clear from the start (ie introduction) that they are proposing a better (ie similar bias, better precision) estimator for the mode than the kernel density estimator of Hartwig et al, and that it similarly relies crucially on ZEMPA.

Response

We agree that our model does imply the ZEMPA assumption. In the revision, we have now explicitly mentioned this in the Introduction (lines 96-98) and further explained why this is true in the Methods (lines 420-429). We further connected MRMix to the mode-based estimator in the Discussion Section in lines 258-260.

Comment

The authors might mention related work of Guo Z et al (JRSS(B) 2018 , discussed in Hartwig et

al) and Burgess et al (IJE 2018, <https://doi.org/10.1093/ije/dyy080>) who have another approach to mode estimation through mixture distributions.

Response

Thank you pointing this reference. Because the method proposed a method for “selection” of valid instruments, in contrast to our which does not perform any such selection based on thresholding, we find the method belongs to a different class. Nevertheless, we find it an interesting method and include it in the discussion in the context of other selection-based estimator such as MRPresso as follows:

A related method for instrument selection, termed two stage hard thresholding (TSHT) with voting, constructs many estimates of the set of valid IVs and use majority or plurality voting to make final decisions. This method was proved to be consistent in instrument selection and effect estimation, but requires individual level data and is not as widely applicable as summary level data based methods. (lines 312-317)

Minor comments:

2. Of course, the simulations were also based on ZEMPA, and the present method would do worse under simulations that favoured MR-Egger or the weighted median. To include such simulations would lengthen the paper unnecessarily, but it would be worth acknowledging the fact.

We acknowledged this fact in lines 363-366.

3. L107, "we do not use the likelihood ... to reduce sensitivity" reads a bit clumsy, suggest rephrasing

We have deleted this sentence (the point has been explained in methods in more details).

4. L128 and throughout, make clear that we assume linear regression models for exposure and outcome - in fact it would be good to write the models out explicitly.

We believe that the key assumption is really the proportionality assumption $\beta_y = \theta\beta_x$, which holds exactly for linear models and approximately for logistic models. We explained in the Methods section (lines 392-403):

The proportional relationship holds if X and Y follow linear models of the form $Y = \alpha_y + \theta X + \epsilon_y$ and $X = \alpha_x + \beta_x G + \epsilon_x$ with the assumption that $E(\epsilon_y|G, X) = E(\epsilon_y|X) = 0$ and $E(\epsilon_x|G) = 0$. Further, the relationship holds for log-linear model for binary Y . To see this, note that if we assume $P(Y = 1|X) = \exp(\alpha_y + \theta X)$ and the regression model for X is $X = \alpha_x + \beta_x G + \epsilon_x$ where ϵ_x is independently distributed of G , then we can derive $P(Y = 1|G) = \exp(\alpha_y) E\{\exp(\theta X) | G\} \propto \exp\{\theta\beta_x G\}$ under the classic instrumental variable assumption that

Y and G are independent given X. Thus, the association between Y and G also follows log-linear model and the underlying association parameter is given by $\beta_y = \theta\beta_x$. The proportionality assumption also approximately holds under logistic regression model when the outcome is relatively rare (in the population) under which log-linear and logistic models become similar. In our data analysis for disease outcomes, we assume that the proportionate relationship holds in the log-odds-ratio parameter scale.

5. L233-4 "overlapping SNPs have strongly correlated effects" - what is the value of this correlation?

The correlation is 0.5 (clarified in line 526)

6. L267 typo, "there would be is an"

Deleted "is" (line 559).

7. L313 the LD score regression method is I think novel in this paper, and nice. It is equivalent to Egger regression over the whole genome, as is clear from the derivation. It would be worth saying this explicitly, in addition to the current note that both methods assume InSIDE. In fact, any method that estimates genetic correlation could be used in this way (GCTA, AVENGEME, ...)

Although it's not widely used, we are not sure if it's novel, since it was discussed in the Online Methods of the bivariate LD score regression paper (Bulik-Sullivan, B. et al. An atlas of genetic correlations across human diseases and traits. *Nat Genet* **47**, 1236-1236 (2015).). We now acknowledge that any method that estimates genetic correlation can be used this way (lines 611-612).

This procedure is nearly equivalent to Egger regression. We stated this in lines 609-610 and supplementary notes section 3).

8. L335 the performance of the mode estimator could probably be improved by adjusting the bandwidth; maybe state this as a limitation of that method.

Added in lines 264-266.

9. L411 typo "indicted" Changed to "indicated" (line 201).

10. l475 type "casual" Changed to "causal" (line 268).

11. L536 "as well as none of the other ... methods" - reads a bit clumsy, please rephrase

We rephrased this sentence (now line 333-335).

12. L572 please delete "unique" (other methods would perform better in situations that

avored their assumptions) Changed to “remarkable” (line 374).

Point-by-point responses to reviewer #2

We thank the reviewer for the valuable comments.

Comment

The authors could provide more details in motivating their mixture modeling methods. Typically the MA analysis starts with genetic variants from existing GWAS for the intermediate phenotypes, so that there is assured association between instrument and the intermediate phenotype. If so, why type (3) and (4) in Figure 1 is ever needed?? The authors need to start with the three classical IV assumptions including non-zero instrumental strength.

Response

We agree with the reviewers that most MR analysis start with SNPs that are established to be associated with X at a genome-wide significance level where the chance of being false positive is extremely low. However, there could be potential for improving power for MR analysis, similar to what has been observed for building polygenic-score for risk-prediction, by including additional SNPs below genome-wide significance threshold. Because we wanted to explore this possibility (see our results in Supplementary Table 5 and Discussion of Results in Line 350-357), we developed the methodology in a broader setting than the usual MR analysis. Results show that more liberal thresholds can improve the efficiency of MRMix estimator, but may lead to winner’s curse bias. The bias disappears when the SNP selection and coefficient estimation are performed based on independent datasets. We would like to explore this possibility further in the future. Allowing for scenarios (3) and (4) gives a broader framework (explained in Methods, lines 417-418).

In fact, we are not the only ones. There are a number of other recent methods, developed for making inference on causal relationship among traits, have considered framework that can allow inclusion of larger set of SNPs, including the whole genome-wide panel, some of which may not be a priori known to be associated with any of the traits. (Zhao Q et al, 2018, <https://arxiv.org/abs/1804.07371>; O’Connor and Price, 2018, <https://www.nature.com/articles/s41588-018-0255-0>)

Comment

the notation in Methods is hard to follow: what is σ^2_x , σ^2_y and σ_{xy} ? define π_1 ... π_4 when they first appear. What is π_0 ? why (2) and (3) collapse in line 160. what is σ^2 in line 173? The derivation of methods in page need clarification/explanation.

Response

We apologize for not being more clear with our notation. We have now tried to add further clarifications as follow:

In legend of Figure 1 we added

“Parameters $\pi_1, \pi_2, \pi_3, \pi_4$ are the mixing probabilities associated with components (1),(2),(3) and (4), respectively; within each component, we assume u_x and u_y to either be 0 or follow normal distributions and $\sigma_x^2, \sigma_y^2, \sigma_{xy}$ are the variance-covariance parameters.”

In Line 467-469, we added

“Under the working model $\hat{\beta}_y - \tilde{\theta}\hat{\beta}_x \sim \pi_0 N(0, s_y^2 + \tilde{\theta}^2 s_x^2) + (1 - \pi_0)N(0, \sigma^2)$, π_0 is the proportion of valid IVs and σ^2 is an unknown variance parameter associated with invalid IVs”.

In Supplementary Notes (bottom of page 1 and top of page 2), we added further clarification why components 2 and 3 collapse as follows:

With θ being the true causal effect, the residual $\beta_y - \theta\beta_x$ is equal to $(u_y + \theta u_x) - \theta u_x = u_y$, which is exactly 0 in scenarios 1 and 4, and has distribution $N(0, \sigma_y^2)$ in scenarios 2 and 3 (Figure 1). Therefore, components 4 collapses with component 1 and components 3 collapses with component 2.

Comment

The data section needs more description about sample size, where the data came from and what conclusion has been draw by previous analyses.

Response

We included this information in Supplementary Table 2, where we reported year published, sample size, and number of genetic loci discovered. We summarized some conclusions from previous MR studies in the Results and Discussion sections:

Lines 180-181: Lipids on coronary artery diseases

Lines 192-196: BMI on breast cancer

Lines 209-212, 272-276: Age at menarche, BMI and breast cancer

Lines 294-296: Year of education on major depressive disorder

REVIEWERS' COMMENTS:

Reviewer #1 (Remarks to the Author):

The authors have addressed most of my comments. I have a few comments outstanding.

1. Although the authors do now note their ZEMPA assumption, they are not quite accurate in doing so. L420 states that the ZEMPA assumption is implied by the four component mixture model. This is incorrect - at this point, no assumption about the distribution of the u_y is needed. It only becomes relevant when fitting the model, ie around L443, because the model fitting assumes that the most common value of u_y is zero. (If all the u_y were say 1, the procedure would be systematically biased). L425 onwards is also not accurate - the ZEMPA assumption is about the mode of u_y , which is well defined even if β_x is zero. I'd suggest redrafting the new paragraph at L420 and the text at L443.

2. The discussion of log linear models at L392 is OK but has been covered before, eg Bowden & Vansteelandt, Stat Med 2011, so this is possibly more detailed than necessary.

3. The title contains an active verb, which is discouraged by this journal.

Reviewer #2 (Remarks to the Author):

The authors addressed my comments

Response to reviewers

Reviewer #1 (Remarks to the Author):

The authors have addressed most of my comments. I have a few comments outstanding.

We thank the reviewer for the comments.

1. Although the authors do now note their ZEMPA assumption, they are not quite accurate in doing so. L420 states that the ZEMPA assumption is implied by the four component mixture model. This is incorrect - at this point, no assumption about the distribution of the u_y is needed. It only becomes relevant when fitting the model, ie around L443, because the model fitting assumes that the most common value of u_y is zero. (If all the u_y were say 1, the procedure would be systematically biased). L425 onwards is also not accurate - the ZEMPA assumption is about the mode of u_y , which is well defined even if β_x is zero. I'd suggest redrafting the new paragraph at L420 and the text at L443.

L420 of last version: We apologize for not being clear about our assumption. In Figure 1a, we also introduced the distributional assumptions that non-zero effects follow normal distributions centered at 0. We added the following sentence to the paragraph above (4th paragraph of Methods – Model setup) to further explain.

“We also make distributional assumptions for the four types of effects: (1) $u_x \sim N(0, \sigma_x^2)$, $u_y = 0$, (2) $\begin{pmatrix} u_x \\ u_y \end{pmatrix} \sim N\left(\begin{pmatrix} 0 \\ 0 \end{pmatrix}, \begin{pmatrix} \sigma_x^2 & \sigma_{xy} \\ \sigma_{xy} & \sigma_y^2 \end{pmatrix}\right)$, (3) $u_x = 0$, $u_y \sim N(0, \sigma_y^2)$, (4) $u_x = u_y = 0$.”

We also updated this paragraph to (5th paragraph of Methods – Model setup):

“Note that our model implies the zero modal pleiotropy (ZEMPA) assumption, which is also required by the mode-based estimator²¹. To see this, note that the pleiotropic effect $u_y = 0$ in the first and fourth components, and follows continuous distribution $N(0, \sigma_y^2)$ in the second and third components. Hence the most common value of u_y is 0, which is equivalent to the ZEMPA assumption.”

(last paragraph of Methods – Model setup, right before MR analysis using mixture-model - MRMix):

2. The discussion of log linear models at L392 is OK but has been covered before, eg Bowden & Vansteelandt, Stat Med 2011, so this is possibly more detailed than necessary.

We shortened this paragraph accordingly and cited the paper Bowden & Vansteelandt, Stat Med 2011 (2nd paragraph of Methods – Model setup).

3. The title contains an active verb, which is discouraged by this journal.

We changed the title to remove the action verb.

Reviewer #2 (Remarks to the Author):

The authors addressed my comments

We thank the reviewer for the comments.